# Mechanisms and Therapeutic Regulation of Pyroptosis in Inflammatory Diseases and Cancer

**DOI:** 10.3390/ijms21041456

**Published:** 2020-02-20

**Authors:** Zhaodi Zheng, Guorong Li

**Affiliations:** Shandong Provincial Key Laboratory of Animal Resistant, School of Life Sciences, Shandong Normal University, Jinan 250014, China; 2016010084@stu.sdnu.edu.cn

**Keywords:** pyroptosis, GSDMD, GSDME, inflammatory disease, cancer therapy

## Abstract

Programmed Cell Death (PCD) is considered to be a pathological form of cell death when mediated by an intracellular program and it balances cell death with survival of normal cells. Pyroptosis, a type of PCD, is induced by the inflammatory caspase cleavage of gasdermin D (GSDMD) and apoptotic caspase cleavage of gasdermin E (GSDME). This review aims to summarize the latest molecular mechanisms about pyroptosis mediated by pore-forming GSDMD and GSDME proteins that permeabilize plasma and mitochondrial membrane activating pyroptosis and apoptosis. We also discuss the potentiality of pyroptosis as a therapeutic target in human diseases. Blockade of pyroptosis by compounds can treat inflammatory disease and pyroptosis activation contributes to cancer therapy.

## 1. Introduction

Many disease states are cross-linked with cell death. The Nomenclature Committee on Cell Death make a series of recommendations to systematically classify cell death [1,2]. Programmed Cell Death (PCD) is mediated by specific cellular mechanisms and some signaling pathways are activated in these processes [3]. Apoptosis, autophagy and programmed necrosis are the three main types of PCD [4], and they may jointly determine the fate of malignant tumor cells. Pyroptosis is a form of programmed necrosis and was firstly described in myeloid cells infected by pathogens or bacteria in 1992 [5,6,7]. It is thought to play a crucial role in the clearance of multiple bacterial and viral infections through removing intracellular replication niches and improving the host’s defensive responses [8]. Pyroptotic death is an inflammatory form of PCD characterized by cellular swelling and rupture, lysis and release of pro-inflammatory molecules such as Interleukin 1β and Interleukin 18 (IL-1β and IL-18) [9,10]. On one hand, pyroptosis has been regarded as exclusive to specialized immune normal cells (such as macrophages, monocytes and dendritic cells) [11,12] and non-immune cell types (such as intestinal epithelial cells, human trophoblasts and hepatocyte cells) because of its inflammatory role [13]. Appropriate inflammatory responses have certain benefits to humans, and the release of cytokines may contribute to tissue angiogenesis [14]. Overactivated pyroptosis can result in a massive inflammatory response leading to inappropriate repair for damaged tissue and organ, causing inflammatory diseases [9,15,16]. However, the long-term exposure of normal tissues to the inflammatory environment may increase the risk of cancers (such as colorectal cancer) [9,15]. It is needed to explore the effect of the blockage of pyroptosis for treating inflammatory factors-driven disease [17,18]. On the other hand, resisting cell death is the important hallmark of cancers [19]. Induction of cancer cell pyroptosis by various stimulations can eliminate malignant cells [20,21,22,23,24]. Although pyroptotic death is often harmful to normal tissues, it can be beneficial to cancer treatment. In recent years, several studies have focused on the relationship between pyroptosis and various human diseases, especially inflammatory diseases and cancer.

In this review, we put forward an overview of how the two gasdermin molecules (D and E) (GSDMD and GSDME) induce pyroptotic death. We summarize and discuss the potential effects of pyroptosis in inflammatory diseases and anticancer therapy.

## 2. Molecular Mechanisms of Pyroptosis

Pyroptosis is induced by members of the gasdermin superfamily, including GSDMA, GSDMB, GSDMC, GSDMD and GSDME [25,26,27,28,29,30], of which, GSDMD and GSDME are widely studied in pyroptosis. These proteins have been shown to have inherent necrotic activity in their gasdermin-N domain, which is usually masked by their gasdermin-C domain [26,29,31]. Proteolytic cleavage between their gasdermin-N and -C domains releases inhibitory gasdermin-C domains, translocating necrotic gasdermin-N domain into the plasma membrane and forming oligomers [26,29,31,32,33,34]. These oligomers form transmembrane pores, allowing the secretion of inflammatory molecules, which disrupt osmotic potential to cause cell swelling with large bubbles blowing from the plasma membrane [27,28,29].

Of gasdermin family numbers, GSDMA, GSDMB and GSDMC proteins have a pore-forming gasdermin-N domain, but they have not been shown to be cleaved to form functional pores in response to physiological or pathological stimuli [25,26]. Only GSDMD and GSDME are cleaved by caspases between their gasdermin-N and -C domains to form membrane pores [25,26,27,28,29,30]. Generally, GSDMD, the downstream of inflammasome activation, is cleaved by inflammatory caspases (caspase1/4/5/11) to induce pyroptosis, while GSDME is cleaved by apoptotic caspase (caspase3) to cause pyroptotic death [26]. Depending on the specific signal pathway and cell types, different molecular patterns are secreted to induce pytoptosis [35].

### 2.1. Mechanism of GSDMD Activation

In the canonical inflammatory pathway, pattern-recognition receptors (PRR) such as Toll-like receptors (TLRs), Nod-like receptor (NLRs) and Absent in melanoma (AIMs), recognize certain pathogen-associated molecular patterns (PAMPs) and certain damaged-associated molecular patterns (DAMPs) [36,37] to activate inflammasomes [38,39,40]. These inflammasomes recruit adaptor protein apoptosis-associated speck-like protein containing a caspase recruitment domain (ASC) to activate caspase1 [41]. Caspase1 can cleave GSDMD to generate the N-terminal domain of GSDMD (GSDMD-N), which permeabilizes the plasma membrane, undergoing pyroptosis [18,42,43,44]. GSDMD has been widely characterized for its ability to form necrotic pores in the plasma membrane. Recently, a study showed that GSDMD causes mitochondrial depolarization and leads to mitochondrial decay before causing plasma membrane rupture upon activation of the inflammasome in macrophages [45]. Rogers et al. further demonstrated that GSDMD-N can translocate to and permeabilize the mitochondrial membrane to activate BCL-2 associated X (Bax) apoptosis regulator and release cytochrome c (Cyt c), triggering the caspase3-mediated mitochondrial apoptotic pathway [46]. This study provides a link between the pyroptotic and apoptotic pathways, in which the inflammatory caspase cleavage of GSDMD targets mitochondria and activates caspase3 to induce apoptosis (Figure 1).

A recent study also demonstrates that when GSDMD expression is too low to cause pyroptosis in certain cell types (neurons, mast cells, fibroblasts), caspase1 can induce apoptosis through the Bid-caspase9-caspase3 axis [47]. In GSDMD-deficient monocytes and macrophages, caspase1 also activates caspase3 and 7 to induce apoptosis [11]. These data suggest that the initiation of pyroptosis or apoptosis induced by caspase1 depends on the expression level of GSDMD.

### 2.2. Mechanism of GSDME Activation

In 2017, Rogers et al. found that GSDME is specifically cleaved by caspase3 to generate its N-terminal fragment (GSDME-N), perforating plasma membrane to induce pyroptosis [29]. In response to chemotherapy drugs, caspase3 cleavage of GSDME drives pyroptosis in GSDME-expressing cells, including normal human primary cells (such as epidermal keratinocytes, placental epithelial cells and umbilical artery smooth muscle cells) and certain cancer cells (such as neuroblastoma, skin melanoma and gastric cancer cells) [21,30,48]. GSDME-negative cells display typical apoptotic death, and GSDME−/− mice are also protected from tissues injuries upon chemotherapy [30]. Subsequently, Zhou et al. revealed that iron-induced oxidative stress triggers pyroptotic death through which Bax recruited to mitochondria stimulates Cyt c release to enhance caspase9 and caspase3 activation, causing cleavage of GSDME in melanoma cells [49].

GSDME has been shown to be an important mitochondrial pore-forming protein. When extrinsic stimuli such as tumor necrosis factor-α (TNFα) bind to death receptors, caspase8 is activated to lead to cleavage of caspase3. Generation of the GSDME-N by active caspase3 can permeabilize the mitochondrial membrane to release Cyt c and induce caspase3-mediated apoptosis, which is similar to the GSDMD-induced apoptotic pathway [46,50].

Above all, GSDME cleavage of apoptotic caspase can target the plasma membrane to drive pyroptosis, and also permeabilize the mitochondrial membrane to augment the mitochondrial apoptotic pathway, both types of cell death share the apoptotic pathway (Figure 2). This discovery alters our understanding of programmed cell death due to the issue that caspase3 has been considered a hallmark of apoptosis all the time. Here, GSDME permeabilization of the plasma or mitochondrial membrane determines the form of cell death in GSDME-activated cells [30].

In addition to the inflammatory caspase1 and apoptotic caspase involved in GSDMD- or GSDME-induced pyroptosis, recent studies show that pyroptosis can be activated by other caspases. Caspase8 activation results in cleavage of both GSDMD and GSDME when Pathogenic Yersinia represses the mitogen-activated protein kinases (MAPK), TGFβ-activated kinase 1 (TAK1), via the effector YopJ, and thereby induces pyroptosis in murine macrophages [51]. In human peripheral blood mononuclear cells (PBMC)-derived macrophages, TAK1 is inhibited by small-molecule 5z7 to block all pro-IL-1β induced by lipopolysaccharide (LPS) stimulation, causing caspase8 activation and GSDME cleavage without the active fragment of GSDMD to induce cell death. It suggests that pro-survival factors in human macrophages are able to bypass TAK1 inhibition to induce less cell death than that in murine macrophages, and thus, human macrophages may have an inborn tolerance to highly pathogenic bacterial neighbors to harm the organism [51]. The differential response of human and murine macrophages shown by these studies [51,52] requires us to explore both mouse and human cells for understanding the course of human disease.

### 2.3. The Association of Pyroptosis and Apoptosis

It is worth noting that apoptosis and pyroptosis can be distinguished by their unique morphological differences and physiological functions [29,48]. However, the underlying mechanisms are complex [53,54]. GSDME is a critical substrate of caspase3 and a key mediator of cell pyroptosis, which occurs after apoptosis [30]. Apoptosis and pyroptosis can be simultaneously regulated by caspase3 or caspase8 due to different cell types or stimulation. In response to TNF or chemotherapy drugs, caspase3 drives pyroptosis in GSDME-high cells and causes apoptosis in GSDME-low cells [30]. Caspase8 activation results in cleavage of both GSDMD and GSDME to drive pyroptosis in murine macrophages, and loss of GSDMD hinders membrane rupture, reverting the cell-death morphology to apoptosis [51]. Since some studies have confirmed that pyroptosis could inhibit the apoptotic pathway in macrophages [11,29], the presence of a pyroptosis pathway may be an alternative way for drugs to kill tumor cells in human glioblastoma multiforme (apoptosis-resistant cancers) [55]. Overall, these reports have shown that apoptosis and pyroptosis are independent but are interconnected through similar pathways and share effector molecules. Further exploration of the mechanisms and relationships of the two pathways may benefit the study of pathological process.

## 3. Roles of Pyroptosis in Inflammatory Diseases

Considering the basic and general functions of pyroptosis under different pathophysiological conditions, the gasdermin family may play an important role in human health and disease. Pyroptosis occurrence leads to the release of pro-inflammatory cytokines IL-1β and IL-18 to the extracellular environment, causing inflammatory effects that contribute to diseases [8]. It is reported that some molecules or compounds which block pyroptosis may lead to effective treatments for various inflammatory diseases (Table 1).

### 3.1. Receptor Protein-Mediated Pyroptosis Effects Inflammatory Diseases

It is well known that pyroptosis plays an important role in pathogenesis of inflammatory diseases, and excessive pyroptosis is often associated with robust inflammation that can result in a number of inflammatory diseases. TNF-α induces inflammatory response to drive pyroptosis by increasing IL-1β, IL-18, caspase1 and nod-like receptor protein 3 (NLRP3) expression in human umbilical vein endothelial cells by activation of the phosphatidylinositol 3-kinase/protein kinase B (PI3K/AKT) pathway. Apolipoprotein-M and Sphingosine-1-Phosphate (S1P) could bind to S1P receptor 2 (S1PR2) to decline TNF-α-induced pyroptosis, which may be associated with attenuation of atherosclerosis [17]. Toll-like receptor 4 (TLR4)/NF-kappaB (NF-κB) is activated by LPS to increase Never in Mitosis A-related kinase 7 (NEK7) expression. NEK7 interacts with NLRP3 to trigger pyroptosis, finally affecting inflammatory bowel disease progression in intestinal epithelial cells and dextran sulfate sodium-induced chronic colitis in mice [66]. Massive hepatocyte death is the core event in acute liver failure (ALF). In liver tissue from ALF patients and a hepatocyte injury mice model, GSDMD-mediated hepatocyte pyroptosis in response to ALF recruits macrophages by increasing protein levels of monocyte chemotactic protein1 (MCP1) and its receptor, CC chemokine receptor-2 (CCR2) to release inflammatory mediators, and finally aggravates ALF [13].

### 3.2. Non-Coding RNA-Regulated Pyroptosis Impacts on Inflammatory Diseases

MicroRNAs (miRNAs) are noncoding single-stranded RNA that can regulate the expression of diverse target genes. High glucose enhances IL-1β-mediated pyroptosis by targeting NLRP1 and activating the nicotinamide adenine dinucleotide phosphate (NADPH) oxidase 4 (NOX4)/ROS/TXNIP/NLRP3 pathway in a cell culture model of diabetic retinopathy (DR). Overexpressed miR-590-3p inhibits pyroptosis and it might be an effective interfering target for the prevention and treatment of DR [42]. Hydrogen peroxide causes cardiomyocytes injury and mice are ligated with the left anterior descending coronary artery to induce myocardial infarction (MI). miR-135b is reduced after cardiomyocytes injury in vivo and in vitro, and pyroptosis is activated. Overexpressed miR-135b can inhibit the NLRP3/caspase1/IL-1β pro-inflammatory pathway and pyroptosis to restore impaired cardiac function [67]. Diabetic corneal endothelial keratopathy is an intractable ocular complication which seriously threatens vision. It has been suggested that diabetes is associated with pyroptosis, long noncoding RNA (lncRNA)KCNQ1 overlapping transcript 1 (KCNQ1OT1) triggers NLRP3 inflammasome activation and caspase1-mediated pyroptosis via targeting miR-214 induced by high glucose in diabetic human and rats’ corneal endothelium [68]. Inflammasome could contribute to ischemic brain injury by inducing inflammation and pyroptosis. LncRNA maternally expressed gene 3 (MEG3) triggers caspase1/GSDMD-mediated pyroptosis through inflammasome absent in melanoma 2 (AIM2) to promote cerebral ischemia-reperfusion injury and induce ischemic stroke in an oxygen-glucose deprivation/reperfusion-treated neurocytes model and in a middle cerebral artery occlusion rat model [69]. Pyroptosis-induced inflammation is involved in the development of human inflammatory diseases. Atorvastatin decreases expression of NLRP3, caspase1, GSDMD and inflammatory factors (IL-1β and IL-18) via the lncRNA nexilin F-actin binding protein antisense RNA 1 (NEXN-AS1)/NEXN pathway to inhibit human vascular endothelial cell pyroptosis, reversing atherosclerosis [56].

### 3.3. Compounds Inhibiting Pyroptosis to Treat Inflammatory Diseases

Emerging research suggests that activation of the NLRP3 inflammasome leads to pyroptosis death, and it is the potential drug target for inflammatory diseases. Isoflurane general anesthesia induces NLRP3 activation, caspase1 cleavage, IL-1β and IL-18 release and the activation of pyroptosis, and consequently causes neuronal damage and cognitive impairment in aged mice. A specific small molecule inhibitor of NLRP3, MCC950, suppresses isoflurane-induced proinflammatory cytokines’secretion to weaken cognitive impairment [57]. NLRP3, overactivated in global and myeloid cell-specific conditional mutant NLRP3 knock-in mice, leads to serious liver inflammation and fibrosis. Anakinra, an IL-1 receptor antagonist, effectively attenuates liver disease by inhibiting the NLRP3/caspase1 pathway [58].

The GSDMD-mediated pyroptosis pathway is closely related to the development of human inflammatory diseases. It may be considered as a potential drug-target to treat inflammatory diseases. Here, several compounds have been identified to repress the GSDMD-associated pathway of pyroptotic death directly or indirectly, attenuating related inflammatory disease. A novel anthraquinone compound named Kanglexin reduces inflammasome activation and subsequent caspase1/GSDMD-mediated pyroptotic death in cardiomyocytes and mice of ligation of coronary artery to induce myocardial infarction, preventing cardiac damages and alleviate ischemic heart disease [59]. Chitosan hydrogel enhances the therapeutic efficacy of bone marrow-derived mesenchymal stem cells for MI by alleviating caspase1/GSDMD-mediated cell pyroptosis in vascular endothelial cells of a mouse model [60]. Ethyl pyruvate markedly suppresses caspase11/GSDMD-mediated pyroptosis induced by LPS and bacterial outer membrane vesicles (OMVs) in macrophage and in the mouse cecal ligation and puncture peritonitis sepsis model, to effectively protect against lethal endotoxemia and rescue sepsis [61]. Huai Qi Huang consists of Chinese medicines of Huaier, Chinese wolfberry and polygonatum sibiricum, and it has been reported to suppress the caspase1/GDSMD-mediated pyroptosis pathway in the joint synovial tissues effectively, and then improve joints’ inflammation of juvenile collagen-induced arthritis rats [63]. Andrographolide [64] and dendrobium alkaloids [65] are the active components respectively extracted from *andrographis paniculate* and Chinese herbal medicine *dendrobium*. In vitro, andrographolide significantly inhibits AIM2 inflammasome and blocks caspase1/GSDMD mediated-pyroptosis in bone marrow-derived macrophages exposed to radiation. When C57BL/6 mice are exposed to whole thorax irradiation and andrographolide is injected intraperitoneally for 4 weeks, the results show that andrographolide significantly delays radiation-induced activation of the AIM2 and pyroptosis in vivo, finally improves radiation-induced lung tissue damage and progressive fibrosis [64]. Pyroptosis occurs after cerebral ischemia injury to activates secretion of inflammatory cytokines IL-1β and IL-18. Dendrobium alkaloids can decrease expression of pyroptosis-related proteins caspase1 and GSDMD, particularly in the hippocampal region of mice to suppress pyroptosis-induced neuronal death, protecting against cerebral ischemia-reperfusion (CIR) impairment [65]. Exposure to hypoxia induces excessive endoplasmic reticulum and unfolded protein response to enhance the NLRP3 inflammatory pathway through thioredoxin-interacting protein (TXNIP) and activate caspase1/GSDMD-mediated pyroptosis in primary human trophoblasts. Resveratrol, a TXNIP inhibitor, decreases the expression of NLRP3 and caspase1 to contribute to therapy of early onset preeclampsia pathology [18]. Clinically, high glucose activates pyroptosis through the caspase1/GSDMD/IL-1β pathway to release proinflammatory cytokines, repressing the proliferation and differentiation of osteoblast in alveolar bone, and Ac-YVAD-CMK, a caspase1 inhibitor, can reverse expression of these proteins and attenuate the formation of periodontal diseases [62]. The expression of caspase1, GSDMD, IL-18 and IL-1β increase significantly in the endothelial cells of subablated hepatic hemangioma in thirty-two patients after radiofrequency ablation, and the caspase-1-associated endothelial pyroptosis is induced by subablative hyperthermia and Ac-YVAD-CMK attenuates pyroptosis in vitro experiments [70].

### 3.4. Other Ways to Regulate Pyroptosis Involved in Inflammatory Diseases

A study reports that mitochondrial aldehyde dehydrogenase 2 (ALDH2) overexpression reduces reactive oxygen species (ROS) production and alleviates high glucose-induced pyroptosis by inhibiting activation of NLRP3 inflammasome and caspase1 in cardiac cells, thereby preventing myocardial injury [71]. Cisplatin is one of the most effective antitumor agents, the clinical use of which is highly limited with its severe side effects, especially the nephrotoxicity. Cisplatin induces GSDMD-mediated pyroptosis in both renal tubular epithelial cells and mouse kidney tissues to increase renal inflammatory cytokine secretion and contribute to acute kidney injury [44]. Visual cycle anomalies induce aberrant build-up of all-trans-retinal (atRAL) in the retinal pigment epithelium (RPE), which result in RPE atrophy in age-related macular degeneration. Furthermore, abnormal accumulation of atRAL activates NLRP3 inflammasome to promote the death of RPE cells due to a significant increase in the proportion of human ARPE-19 cells exhibiting features of caspase3/GSDME-mediated pyroptosis [72]. Sodium butyrate breaks down cell–cell junctions and triggers caspase3/GSDME-dependent pyroptosis in the gingival epithelial cells to destroy the epithelial barrier and shed a new light on our understanding of periodontitis initiation [73]. GSDME-mediated pyroptosis is poorly studied in inflammatory disease. Therefore, more experiments are needed to investigate the potential mechanism and application of inflammatory disease based on GSDME-induced pyroptosis.

## 4. Inducing Cancer Cell Pyroptosis for Cancer Therapy

Under certain circumstances, cell death is undoubtedly beneficial to human health, specifically to cancer therapy [74]. Pyroptosis, as a type of cell death, can suppress the occurrence and development of cancers [75]. Some studies indicate that chemotherapy drugs, reagents, natural products and target therapy drugs could trigger pyroptosis in various types of cancer (Table 2). Thus, the pyroptotic death will be a new target in cancer therapy.

### 4.1. Chemotherapy Drugs-Induced Pyroptosis Possesses Anticancer Effects

Chemotherapy drugs are known to elicit cell death by activating pyroptosis. Shao et al. originally proposed that chemotherapy drugs such as doxorubicin, actinomycin-D, bleomycin and topotecan can induce pyroptosis through caspase3 cleavage of GSDME in lung cancer NCI-H522 cells [30]. Doxorubicin triggers GSDME-dependent pyroptosis in melanoma cell lines with high expression of GSDME. Suppression of elongation factor-2 kinase (eEF-2K) leads to autophagy inhibition and pyroptosis enlargement to reinforce the antitumor efficacy of doxorubicin to melanoma cells [76]. Lobaplatin causes ROS elevation and c-jun n-terminal kinase (JNK) activation to drive caspase3/GSDME-mediated pyroptosis by activating the mitochondrial apoptotic pathway in colon cancer cells [22]. More recently, both chemotherapeutic cisplatin and paclitaxel induce pyroptosis via caspase3/GSDME activation; however, cisplatin triggers the pyroptosis more potently than paclitaxel in lung cancer A549 cells [21]. These data provide the theoretical basis for the application of pyroptosis when chemotherapy drugs are exposed to cancer cells, and further indicate that different chemotherapeutic agents show different capability of pyroptosis induction in cancer celllines with GSDME expression.

The chemotherapy combination with other ways more efficiently promotes pyroptosis of cancer cells to stimulate intense immune responses for preventing from tumorigenesis than monotherapy. For example, a polo-like Kinase 1 (PLK1) inhibitor could sensitize esophageal squamous cell carcinoma (ESCC) cells to cisplatin by inhibiting the DNA damage repair pathway and inducing Bax/caspase3/GSDME pathway-mediated pyroptosis. The new information indicates that PLK1 inhibitor may be an attractive candidate for ESCC targeted therapy, especially when combined with cisplatin for treating GSDME-overexpressing tumors [77]. In a recent study, Fan et al. developed a strategy to combine decitabine (DAC) with chemotherapy nanomedicines, triggering tumor cell pyrolysis through epigenetics, thereby further enhancing the immunological effects of chemotherapy. DAC is performed in advance using specific tumor-bearing mice to demethylate the *GSDME* gene in mouse colon cancer cells and breast carcinoma cells. Subsequently, the tumor-targeting nanoliposomes loaded with cisplatin (LipoDDP) is used to manage drugs that activate the caspase3 pathway in tumor cells and trigger pyroptosis. These findings suggest that DAC can be considered a pretreatment adjuvant in combination with chemotherapy to promote the development of tumor cell pyroptosis through caspase3. By reversing GSDME expression in tumor cells with DAC pretreatment, LipoDDP is ready to deliver chemotherapy drugs targeting mice tumor sites to prevent normal tissues from side effects [78]. These experiments reveal the realization and usability of the combination therapy and the cytokine-stimulated immune response during the pyrolysis process, which greatly reduces the recurrence after chemotherapy.

### 4.2. Non-Chemotherapy Drug-Induced Pyroptosis Exerts Anticancer Effects

High doses of chemotherapeutic drugs can be used to maintain therapeutic activity, but cause adverse reactions, including tissue damage and weight loss [30]. Compound L61H10, a heterocyclic ketone derivative, has exerted the cancer inhibitory effects without obvious side effects both in lung cancer cells and in the nude mice bearing xenografts by arresting the cell cycle in the G2/M phase and mediating the switch of NF-κB-modulated apoptosis to caspase3/GSDME-mediated pyroptosis [79]. Wang et al. showed that metformin, a widely used anti-diabetic drug, is able to activate the GSDMD-mediated pyroptosis of ESCC by targeting the miR-497/Proline-, glutamic acid- and leucine-rich protein-1 (PELP1) pathway to treat ESCC [80]. Many studies have aimed to determine how to maintain therapeutic Arsenic trioxide (As2O3) concentrations in target solid tumor tissues for long period of time via activation of pyroptosis with few side effects [81,86,87]. Local drug delivery systems can extend the retention time of drugs’ administration at the dosing site, resulting in more continuous efficacy and reduced side effects of normal tissues and organs [88,89]. Arsenic trioxide nanoparticles (As2O3-NPs) are prepared via a nano-drug delivery system loading with arsenic trioxide. It induces more increased GSDME-N expression and pyroptosis induction compared with As2O3 in hepatocellular carcinoma (HCC) and Huh7 xenograft-bearing mice [81].

Natural products are widely used for anticancer effects due to their low toxicity, low price, wide source and reduction of drug resistance produced from tumor cells. Both galangin (GG) and anthocyanin widely exist in plants and belong to natural flavonoids. The GG elicits a potent antitumor activity by inducing pyroptosis with activation of caspase3/GSDME, and autophagy inhibition by repressing LC3B enhances GG-induced pyroptosis in glioblastoma cells [55]. Yue et al. found that anthocyanin increases expression of NLRP3 and caspase1 to activate GSDMD-mediated pyroptosis, and subsequently suppresses survival rate and migration and invasion of oral squamous cell carcinoma (OSCC) [82]. Dioscin also induces GSDME-dependent pyroptosis to inhibit the growth of human osteosarcoma [83]. Berberine induces pyroptosis by activating caspase1 to inhibit the viability, migration and invasion capacity of HCC [84]. Huaier extract exhibits an antitumor effect through promoting NLRP3-dependent pyroptotic cell death in non-small cell lung cancer (NSCLC) cells and NSCLC patients [85].

KRAS is an oncogene, and epidermal growth factor receptor (EGFR) and anaplastic lymphoma kinase (ALK) are the drivers of tumorigenesis. A recent study showed that robust pyroptosis is triggered when diverse small-molecule inhibitors specifically target KRAS, EGFR or ALK in lung cancer. Upon treatment of inhibitors, the mitochondrial apoptotic pathway engages and executes caspase3/GSDME-induced pyroptosis [90]. Similar to *KRAS*, both B-Raf proto-oncogene (*BRAF*) and mitogen-activated protein kinase (*MEK*) are two oncogenes. Combinations of BRAF inhibitors and MEK inhibitors (BRAFi + MEKi) are Food and Drug Administration (FDA)-approved to treat BRAF V600E/K mutant melanoma. BRAFi + MEKi treatment promotes cleavage of GSDME and release of high-mobility group protein B1 (HMGB1) to induce cell pyroptotic death [91]. Transcription factor p53 overexpression triggers pyroptosis to suppress tumor growth in A549 tumor-bearing mice. In clinical trials, p53 expression level is positively related to pyroptosis in tumor tissue of NSCLC patients, implicating the potentiality of p53 on antitumor via induction of pyroptosis [20]. The studies provide new ideas to support pyroptosis as a novel mechanism of molecular-targeted drugs to exterminate oncogene-addicted tumor cells.

## 5. Conclusions and Future Perspectives

Pyroptosis is a new form of programmed cell death and has recently been extensively studied in various diseases. The research done to utilize this pathway for regulation of tissue development and homeostasis has also been highlighted. We concluded the insights into molecular mechanisms of pyroptosis, which pave new ways for disease therapy. Inhibition or activation of pyroptosis have opposing roles in therapy of inflammatory disease and cancer. On one hand, active pyroptosis can be highly pathological and cause human inflammatory diseases, and the blockage of the pyroptosis pathway is used for anti-inflammatory therapy in non-cancerous cells and tissues. On the other hand, less pyroptosis occurs in cancer cells due to low expression of gasdermin and it does not induce cancer cell death. Pharmacological activation of pyroptosis eliminates malignant tumor cells and has become essential for treatment of cancers. However, the roles of pyroptosis in inflammatory disease and cancer research are only just beginning to be comprehended, and further studies of the signal pathways about pyroptosis occurrence need to be investigated to provide new directions for the treatment of inflammatory disease or cancer. Some compounds can act as the promising therapeutic drugs for blockage of pyroptosis in inflammatory disease, and others can induce pyroptosis for cancer therapy. More clinical trials are required to explore the potential application of inflammatory disease and cancer therapy based on pyroptosis.

## Figures and Tables

**Figure 1 ijms-21-01456-f001:**
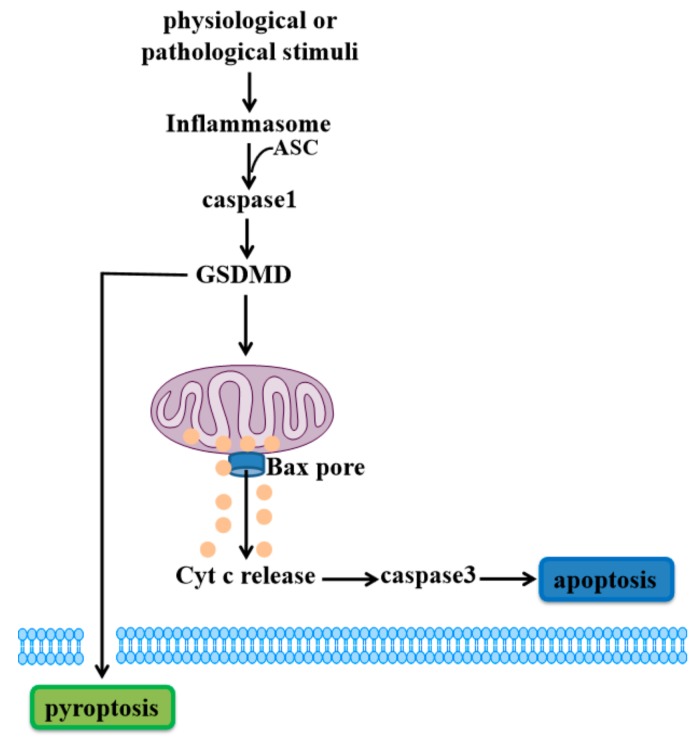
Gasdermin D (GSDMD)-mediated cell death. When physiological or pathological stimuli activates inflammasomes, ASC is recruited to lead to activation of caspase1. GSDMD is cleaved by caspase1 to release active GSDMD-N, translocating to the plasma membrane to induce pyroptosis. Meanwhile, GSDMD-N can also permeabilize the mitochondrial membrane to trigger the mitochondrial apoptotic pathway downstream of inflammasome activation, inducing cell apoptosis.

**Figure 2 ijms-21-01456-f002:**
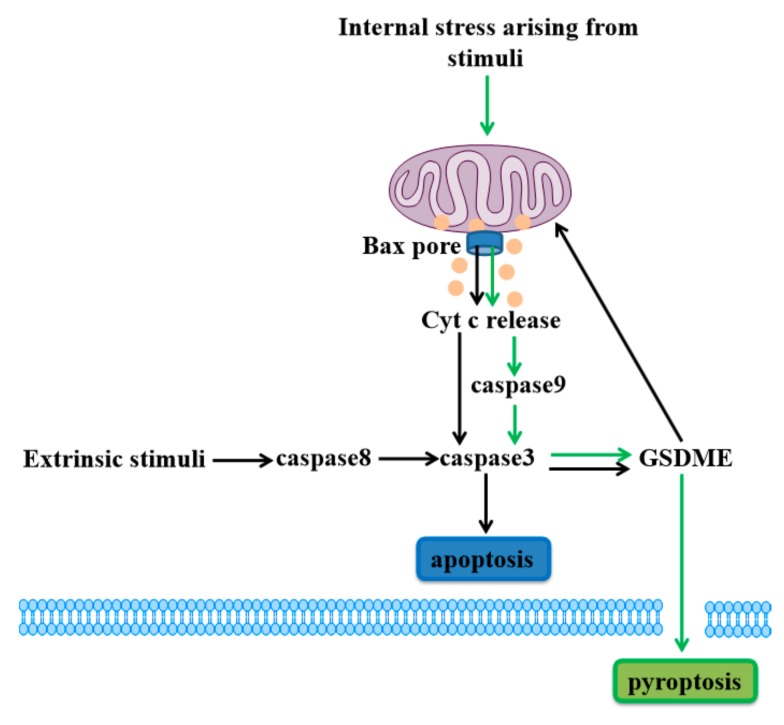
Gasdermin E (GSDME)-dependent cell death. Internal stress arising from stimuli (such as iron) lead to activation of Bax and release of Cyt c to stimulate caspase9 activation and induce caspase3/GSDME-mediated cell pyroptosis (green arrows). Meanwhile, the extrinsic stimuli activate caspase8 and caspase3 to cleave GSDME, and GSDME permeabilizes the mitochondrial membrane to release Cyt c, inducing caspase3-mediated apoptosis (black arrows).

**Table 1 ijms-21-01456-t001:** Compounds inhibiting pyroptosis signal pathway to treat inflammatory diseases.

Classification	Compounds	Inflammatory Diseases	Mechanisms of Pyroptosis Inhibition	References
Reagents	Atorvastatin	Atherosclerosis	lncRNA NEXN-AS1/NEXN /NLRP3/caspase1/GSDMD	[56]
	MCC950	Cognitive impairment	NLRP3/caspase1	[57]
	Anakinra	Liver inflammation	NLRP3/caspase1	[58]
	Kanglexin	Ischemic heart disease	NLRP3/caspase1/GSDMD	[59]
	Chitosan hydrogel	Myocardial Infarction	Caspase1/GSDMD	[60]
	Ethyl pyruvate	Endotoxemia and sepsis	Caspase11/GSDMD	[61]
	Ac-YVAD-CMK	Periodontal diseases	Caspase1/GSDMD/IL-1β	[62]
Natural products	Huai Qi Huang	Arthritis	Caspase1/GDSMD	[63]
	Andrographolie	Lung injury	AIM2/caspase1/GSDMD	[64]
	Dendrobium alkaloids	CIR impairment	Caspase1/ GSDMD	[65]
	Resveratrol	Preeclampsia	TXNIP/NLRP3/caspase1 /GSDMD	[18]

CIR: cerebral ischemia-reperfusion. NEXN-AS1: nexilin F-actin binding protein antisense RNA 1. NLRP3: NOD-like receptor protein 3. GSDMD: gasdermin D.IL-1β: Interleukin1β.AIMs: absent in melanoma like receptors.TXNIP: thioredoxin-interacting protein.

**Table 2 ijms-21-01456-t002:** Compounds inducing pyroptosis signal pathways in related cancers.

Classification	Compounds	Cancer types	Mechanisms of Pyroptosis Induction	References
Chemotherapy drugs	Doxorubicin, Actinomycin-D, Bleomycin, Topotecan	Lung cancer	Caspase3/GSDME	[30]
	Doxorubicin	Melanoma	eEF-2K/caspase3/ GSDME	[76]
	Lobaplatin	Colon cancer	ROS/JNK/caspase3 /GSDME	[22]
	Cisplatin, Paclitaxel	Lung cancer	Caspase3/GSDME	[21]
	Cisplatin+PLK1 inhibitor	ESCC	Bax/caspase3/GSDME	[77]
	Cisplatin + Decitabine	Colon cancer and Breast carcinoma	Caspase3/GSDME	[78]
Reagents	L61H10	Lung cancer	NF-κB/caspase3/GSDME	[79]
	Metformin	ESCC	miR-497/PELP1/caspase1 /GSDMD	[80]
	As2O3-NPs	HCC	Caspase3/GSDME	[81]
Natural products	Galangin	Glioblastoma	LC3B/caspase3/GSDME	[55]
	Anthocyanin	OSCC	NLRP3/caspase1/GSDMD	[82]
	Dioscin	Osteosarcoma	Caspase3/GSDME	[83]
	Berberine	HCC	Caspase1	[84]
	Huaier	NSCLC	NLRP3/caspase1	[85]

ESCC: esophageal squamous cell carcinoma. OSCC: oral squamous cell carcinoma. HCC: hepatocellular carcinoma. NSCLC: non-small cell lung cancer. GSDME: gasderminE. EGFR: epidermal growth factor receptor. ROS: reactive oxygen species. JNK: c-jun n-terminal kinase.Bax: BCL-2 associated X. PELP1: Proline-, glutamic acid- and leucine-rich protein-1.

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
