# Peer review of "Mechanisms and Therapeutic Regulation of Pyroptosis in Inflammatory Diseases and Cancer"

_ijms, 2020, doi:10.3390/ijms21041456_

Round 1

Reviewer 1 Report

In my opinion, several major and minor changes (as shown below) are necessary to improve this review article that actually is unacceptable for publication. 

TITLE-Please change as "Mechanisms and Therapeutic Regulation of Pyroptosis in Inflammatory Diseases and Cancer" Abstract-Line 13 The authors should rewrite the sentence as "This review aims to summarize the latest molecular mechanisms about pyroptosis mediated by pore-forming GSDMD and GSDME proteins that permeabilize plasma and mitochondrial membrane activating pyroptosis and apoptosis".line 16 rewrite as " we also discuss the potentiality of pyroptosis as a therapeutic target in human diseases". INTRODUCTION-Line 24 Please insert -..." cellular mechanisms, linked to activation of signaling pathways [3]." Line 33 Insert.."specialized immune normal cells..". Line 39 the sentence starts as "However, the long-term exposure.." and delete.."And". Line 41 Please rewrite the sentence as " It is needed to explore the effect of the blockage of pyroptosis for treating inflammatory factors-driven disease." and insert Reference here to substantiate this sentence. Line 45 The authors should change the text according to " In recent years, several studies have focused on the relationship between pyroptosis and various human diseases, especially inflammatory diseases and cancer." Molecular mechanisms of pyroptosis-Line 64 Insert References at the end of the sentence.." to physiological or pathological stimuli." Line 70 Instead of "that" insert "by which two key.." Mechanism of GSDMD activation-Line 74 Write "recognize" instead of "recognizes" Line 86 Please rewrite all sentence that sounds not clear to me. Line 91 The authors must follow the proper enumeration of References and not insert 46 here! Moreover, at the end of the sentence, they should add more text to explain the role of caspase 1 in apoptosis. Mechanism of GSDME activation-Line 102 Please the authors should start with "In 2017, Rogers et al..".At Line 107 insert more References. Figure 2 please explain better the meaning or insert more explicative sentences in the text. "2.3 Other mechanisms of GSDMD and GSDME subtitle must be deleted. Line 145 Put References at the end of the sentence..."..physiological functions." However, in my opinion the subsection is too brief and generic and needs more insights from the authors. Line 164-NLRP3 must be written in full words and then abbreviated inside brackets as "(NLRP3)". The same criticism for line 170-S1PR2, Line 172 NEK7, Line 177 KCNQ10T1 et so on.. Lines 215-218-all these sentences must be inserted at the beginning of the subsection 3.3. Line 235-Line 242 herbal or fungi derivatives from traditional Chinese medicine must be also indicated according to the official botanical nomenclature. Line 278 Table 1 should be inserted not here but at the end of subsection 3.2. References must be checked according to the enumeration in the text. Lines 324-325 Please the authors must insert proper References and best specify which the main side effects of chemotherapy are. Lines 352-353 the sentence is in the wrong place, please eventually insert at the end of the text. However, all oncogenes must be explained. The authors are requested to put an "Abbreviations List" in alphabetic order at the end of the text. CONCLUSIONS and future perspectives-In my opinion, the authors must better highlight the opposing role of inhibition or activation of pyroptosis in inflammation and cancer. The authors should include limitations of the study and eventually a summary plot on the importance of pyroptosis regulation. 

Reviewer 2 Report

The review presented by Zheng and Li addresses a topic of great interest, pyroptosis, a death pathway associated with inflammatory phenomena and the role of this process in diseases and cancer therapy.
The first part of review is fluent and pleasant to read, while the second one reviews in a somewhat fragmented and sometimes confused. It may be more linear to gather the results by action mode (for example gather all the examples in which non-coding RNA are involved) or by pathologies.

The bibliography is, on the whole, updated and adequate.
The English language needs a moderate revision: many sentences appear fragmented and many grammatical errors are present.
only as examples:
lines 70-71 Here, we describe the mechanisms that two key molecular players (GSDMD 70 and GSDME) used to trigger pyroptotic death;
line 80 for its ability
line 86-87 two pathway that inflammatory caspase cleaves ...

line 109 Zhou et al.

linea 116-119 the sentence is fragmented

line 129 GSDMD and GSDME action

line 130 caspase involved in 

line 162 It is reported and molecules or compounds which block pyroptosis 

line 181 drug target

line 191 Overexpressed 

line 219 is involved in the development

line 284 There are Some studies 

line 297 More recently 

lines 299-302 sentence fragmented

line 303 efficiently promotes 

line 309 GSDME-overexpressing tumors...

Author Response

Response: Thank you for your comments and valuable suggestions. The line number below is in the present version.
According to your suggestions, we have arranged the second one reviews again, and made more linear to gather the results by action mode (the detail in lines 179-297). The subtitles of this part as following:
3.1 Receptor protein-mediated pyroptosis effects inflammatory diseases
3.2 Non-coding RNA-regulated pyroptosis impacts inflammatory diseases
3.3 Compound inhibition of pyroptosis treats inflammatory diseases
3.4 Other ways-regulated pyroptosis involves in inflammatory diseases

We have made all modifications in green font in the text. The line number of modifications in English language as following: lines 72-73; lines 82-83; lines 88-90; line 113; lines 123-125; line 138; line 139; line 176; line 203; line 219; line 226; line 302; line 314; lines 317-320; line 321; line 327.
Additionally, we have made following modifications in blue font.
In lines 59 and 64, we have changed "domains" with "domain";
In line 151, we have changed " to understand" with "for understanding";
In line 175, we have changed "contributes" with "contribute";
In line 187, we have changed "and" with ", which".

Yours sincerely,
Guorong Li, PhD, Professor
School of Life Sciences,
Shandong Normal University,
E-mail: [email protected]

Round 2

Reviewer 1 Report

The authors ameliorated the manuscript that still deserves to be checked better for typing errors and the English language before acceptance.

 For example REWRITE AS FOLLOWS

Line 198 3.2 Non-coding RNA-regulated pyroptosis impacts on inflammatory diseases

Line 224 3.3 Compounds inhibiting pyroptosis to treat inflammatory diseases

Line 279 3.4 Other ways-regulated pyroptosis involved in inflammatory diseases

Line 387 Table 2. Compounds inducing pyroptosis signal pathways in related cancers

Author Response

Reviewer #1 (Round 2):
The authors ameliorated the manuscript that still deserves to be checked better for typing errors and the English language before acceptance.

For example REWRITE AS FOLLOWS
Line 198 3.2 Non-coding RNA-regulated pyroptosis impacts on inflammatory diseases
Line 224 3.3 Compounds inhibiting pyroptosis to treat inflammatory diseases
Line 279 3.4 Other ways-regulated pyroptosis involved in inflammatory diseases
Line 387 Table 2. Compounds inducing pyroptosis signal pathways in related cancers

Response: Thank you for your suggestions. we have made the modifications in purple font in the text. The line number of modifications in English language as following:
Line 198; line 224; line 279; line 387.
Additionally, in line 277, we have changed "Table 1. Compounds inhibition of pyroptosis signal pathway to treat inflammatory diseases" with "Table 1. Compounds inhibiting pyroptosis signal pathway to treat inflammatory diseases".

Yours sincerely,
Guorong Li, PhD, Professor
School of Life Sciences,
Shandong Normal University,
E-mail: [email protected]
